

# Land-based wind turbines with flexible rail transportable blades – Part II: 3D FEM design optimization of the rotor blades

Ernesto Camarena[1], Evan Anderson[1], Josh Paquette[1], Pietro Bortolotti[2], Roland Feil[2], and Nick Johnson[2]

[1]Sandia National Laboratories, Albuquerque, NM 87185, USA
[2]National Renewable Energy Laboratory, National Wind Technology Center, Golden, CO 80401, USA

**Correspondence:** Ernesto Camarena (ecamare@sandia.gov)

**Abstract.** Increasing growth in land-based wind turbine blades to enable higher machine capacities and capacity factors is creating challenges in design, manufacturing, logistics, and operation. Enabling further blade growth will require technology innovation. An emerging solution to overcome logistics constraints is to segment the blades spanwise and chordwise, which is effective, but the additional field-assembled joints result in added mass and loads, as well as increased reliability concerns

in operation. An alternative to this methodology is to design slender flexible blades that can be shipped on rail lines by flexing during transport. However, the increased flexibility is challenging to accommodate with a typical glass-fiber, upwind design. In a two-part paper series, several design options are evaluated to enable slender flexible blades: downwind machines, optimized carbon fiber, and active aerodynamic controls. Part 1 presents the system-level optimization of the rotor variants as compared to conventional and segmented baselines, with a low-fidelity representation of the blades. The present work,

Part 2, supplements the system-level optimization in Part 1 with high-fidelity blade structural optimization to ensure that the designs are at feasible optima with respect to material strength and fatigue limits, as well as global stability and structural dynamics constraints. To accommodate the requirements of the design process, a new version of the Numerical Manufacturing And Design (NuMAD) code has been developed and released. The code now supports laminate-level blade optimization and an interface to the International Energy Agency Wind Task 37 blade ontology. Transporting long, flexible blades via

controlled flapwise bending is found to be a viable approach for blades up to 100 m. The results confirm that blade mass can be substantially reduced by going either to a downwind design or to a highly coned and tilted upwind design. A discussion of active and inactive constraints consisting of material rupture, fatigue damage, buckling, deflection, and resonant frequencies is presented. An analysis of driving load cases revealed that the downwind designs are dominated by loads from sudden, abrupt events like gusts rather than fatigue. Finally, an analysis of carbon fiber spar caps for downwind machines finds that, compared

to typical carbon fibers, the use of a new heavy-tow carbon fiber in the spar caps is found to yield between 9 % and 13 % cost savings.





manuscript, or allow others to do so, for United States Government purposes. The Department of Energy will provide public access to these re-
sults of federally sponsored research in accordance with the DOE Public Access Plan (http://energy.gov/downloads/doe-public-access-plan).

# 1   Introduction

Wind turbine rotors have been growing at a faster rate than generator capacity, enabling higher energy capture in low and
moderate wind speeds (Bolinger et al., 2020). Thus, the ratio of the nameplate capacity to the swept area, or specific power (W
m$^{-2}$) is diminishing. Using low specific power turbines has increased wind plant capacity factors which is a measure of how
often maximum power is produced. This trend of growing capacity factors (i.e., lower specific power) allows modern turbines
to both produce power more consistently and operate in low wind speed sites. Current blades are approaching 80 m in length
for land-based installations and over 100 m for offshore turbines. Bolinger et al. (2020) has shown that the ongoing reductions
in specific power are likely to continue. While not as much of an issue for offshore wind turbines, land-based machines are
currently constrained by transportation logistics. Current land-based transportation constraints limit monolithic blade lengths
to about 80 m in length, and 4.75 m in width and height. Thus, continued growth in rotor sizes for land-based turbines will
likely require design changes.

In 2018, the U.S. Department of Energy (DOE) funded the Big Adaptive Rotor (BAR) project to study the design drivers
of future high capacity factor land-based turbines, and investigate potential technology solutions to the challenges that these
designs create. The project used a 5 MW, 206 m rotor as a reference turbine platform to examine the limits of current design
tools and methodologies. After an initial study and down-select of turbine innovations (Johnson et al., 2019), the concept
of slender flexible blades that can be transported by train via controlled flapwise bending (Carron and Bortolotti, 2020) was
chosen as the primary concept to be evaluated. Three enabling technologies were selected as having high potential to enable
this concept: downwind rotors, carbon fiber, and distributed aerodynamic controls. Downwind rotors, while having noise and
cyclic loading concerns, experience their highest deflections away from the tower, lessening that constraint. Carbon fiber, while
expensive in its current form, can be used in blades to enable thinner airfoils to allow higher flexibility while maintaining the
required structural integrity. Reference rotor models were designed and optimized for each of these technologies. Each design
is listed below as well as the nomenclature adopted.

– BAR-UAG (Upwind – Air transport – Glass fiber spar caps): A baseline upwind design composed primarily of fiberglass
50       composites with 4 m of blade prebend in a manner similar to current industry standard designs.

– BAR-DRG (Downwind – Rail transportable – Glass fiber spar caps): A straight, slender, and downwind design composed
primarily of fiberglass composites, intended to be transportable by train on existing railways.

– BAR-DRC (Downwind – Rail transportable – Carbon fiber spar caps): A straight, slender, and downwind design using
industry-standard carbon fiber composite for the spar caps, intended to be transportable by train on existing railways.

– BAR-USC (Upwind – Segmented – Carbon fiber spar caps): A segmented upwind design, composed of two sections
using industry-standard carbon fiber composite for the spar caps and attached rigidly together by a mechanical joint.





- BAR-URC (Upwind – Rail transportable – Carbon fiber spar caps): An upwind design using industry-standard carbon fiber composite for the spar caps, intended to be transportable by train on existing railways and installed with 8 degrees of nacelle tilt and 4 degrees of rotor precone, and no blade prebend.

All blades had a total length, $L$, of 100 m. Further detail of the BAR concepts and initial design optimization is found in Part 1 (Bortolotti et al., 2021). Since the use of traditional, aerospace carbon fibers has contributed limited benefit to wind turbines due to high cost, a new low-cost carbon fiber (Ennis et al., 2019), referred to here as *heavy-tow* (HT) carbon fiber, is also evaluated. Thus, the designs with carbon fiber spar caps have the following additional cases:

- BAR-DRCHT: A variation of the BAR-DRC design, using heavy-tow carbon fiber composite for spar caps instead of the
industry standard.

- BAR-USCHT: A variation of the BAR-USC design, using heavy-tow carbon fiber composite for spar caps instead of the industry standard.

- BAR-URCHT: A variation of the BAR-URC design, using heavy-tow carbon fiber composite for spar caps instead of the industry standard.

Optimal designs were obtained by performing numerous iterations between two levels of optimization: each with differing structural fidelities. The National Renewable Energy Laboratory (NREL) led the lower-fidelity system optimization, wherein the levelized cost of energy was minimized while capturing the overall behavior of the system at the turbine level. This was performed first, whereby a preliminary blade design was created. Further details are presented in Part 1 of this two-part series (Bortolotti et al., 2021).

Beam models, in part, enabled the design of the aerodynamic shape and sizing of the spar cap while accounting for stochastic wind fields and the interaction of numerous turbine components. These types of models are used at the expense of accurately quantifying ultimate and fatigue failure of the blades, as well as buckling instabilities. Some researchers have accounted for buckling instabilities with beams by using simple analytical buckling formulas (buckling is a form of instability) (Ning and Petch, 2016; Bir, 2001). These formulas, however, are often derived for flat, homogeneous, and isotropic panels with unrealistic
boundary conditions (i.e., fixed, free, pinned, etc); neither of these conditions apply to most panels in a given blade. Inadequate stress recovery for beam models has also been shown due to blade taper (Bertolini et al., 2019) effects and localized effects near the root are not able to be well-resolved. Note that linear cross-sectional analysis tools (Feil et al., 2020) were utilized in the lower-fidelity optimization. Nonlinear cross-sectional approaches exist (Harursampath and Hodges, 1999; Couturier and Krenk, 2016) and can account for flattening of the cross section under flexure, known as the Brazier effect. These, however,
are limited to standard structural cross sections and will not detect localized buckling of a panel without simplifications.

In an effort to ensure the BAR designs were neither over or underdesigned, the system-level optimization was followed by a high-fidelity structural optimization. The increase in fidelity required keeping the aerodynamic shape unaltered and relying on the lower-fidelity model for the loading information. The present contribution details the high-fidelity methods and associated failure modes employed by Sandia National Laboratories (SNL). It differs from a related work (Bottasso et al., 2014)





in that the structural design characteristics of long, slender wind turbine blades for large, land-based rotors are revealed. Solid finite elements (FEs) with a layer-wise discretization can provide the best approximation to the 3D elasticity boundary value problem. This is because no simplifications or assumptions on the kinematics, local fields, and constitutive response are required. Designing a blade with a model that exclusively uses solid elements would require far too many degrees of freedom due to the vast separation of scales between layer thicknesses and the overall blade size. Even if layers were grouped with a

homogenization theory, the degree of freedom requirement would still be too high for practical design. Shell elements (ANSYS SHELL181) were exclusively utilized in this work because it is a good compromise between fidelity and computational cost.

This design approach required numerous improvements to the MATLAB® code named Numerical Manufacturing And Design (NuMAD) 2.0 (Berg and Resor, 2012). NuMAD 2.0 had a strong dependency on a graphical user interface (GUI), which interferes with optimization. This culminated in the current release of NuMAD; NuMAD 3.0. The new release incorporates

optimization, added structural analyses, and capability to accept input from the International Energy Agency Wind Task 37 blade ontology (referred to here as a *yaml file* (Bortolotti et al., 2019)). It includes the GUI but with the addition of an object-oriented approach. This was done in order to automate the new optimization process, which utilizes MATLAB® gradient-based optimization methods and various calls to ANSYS® to perform meshing and structural analyses. The code can be obtained from https://github.com/sandialabs.

The remainder of this paper summarizes the methods used to derive the material properties needed in the structural analysis and the application of aeroelastic loading to a high-fidelity structural model, and provides details regarding the structural optimization of the wind turbine blade. Specifically, Sect. 2 details how a complete set of material properties required for the optimization and analysis were obtained. Section 3 – 4 detail the updates to NuMAD. Section 3 shows how the loading information was transferred from the lower-fidelity aeroelastic simulations to the higher-fidelity case. Section 4 describes how

the high-fidelity structural optimization was performed. Finally, results are presented in Sect. 5 and conclusions are made in Sect. 6.

## 2 Material Properties

A common difficulty for the composites simulation community is that publicly available experimental data sets often provide a limited subset of inputs needed for accurate modeling. Experimental data for only a unidirectional glass composite was found

to have been fully characterized experimentally with elastic, rupture, fatigue, and mass properties (Samborsky et al.). The rest of the composites required micromechanical analyses and approximate rupture calculations for laminates. This section elaborates on the assumptions and calculations adopted to generate the material properties. The required properties for the composite laminates are shown in Table 1 and 2. Note that $\gamma_\sigma$ is the factor of safety for material rupture.

The uniaxial glass laminate was assumed to comprise Vectorply E-LT-5500 with Epikote MGS RIMR 135/Epicure RIMH

1366 epoxy resin from Samborsky et al.. The biaxal glass composite was assumed to comprise $[\pm 45°]_6$ PPG-Devold DB810-E05—a fabric infused with Epikote MGS RIMR 135/Epicure RIMH 1366 epoxy resin. The triaxial glass composite was



**Table 1.** Mechanical properties of composite laminates by fiber type. All strength properties are design values (not the characteristic values) and are obtained with $\gamma_\sigma = 1.74$.

| Fiber Type | Uniaxial Glass | Biaxial Glass | Triaxial Glass | Uniaxial Carbon | Uniaxial Heavy-Tow Carbon |
|---|---|---|---|---|---|
| $E_1^*$ [GPa] | 43.70 | 11.02 | 28.21 | 157.6 | 160.6 |
| $E_2^*$ [GPa] | 16.50 | 11.02 | 16.24 | 9.1[a] | 9.1[a] |
| $E_3^*$ [GPa] | 15.45 | 16.05 | 15.84 | 9.1[a] | 9.1[a] |
| $G_{12}^*$ [GPa] | 3.265 | 13.23 | 8.248 | 4.131 | 4.131 |
| $G_{13}^*$ [GPa] | 3.495 | 3.488 | 3.491 | 4.131 | 4.131 |
| $G_{23}^*$ [GPa] | 3.480 | 3.488 | 3.491 | 2.689 | 2.689 |
| $\nu_{12}^*$ [ ] | 0.262 | 0.6881 | 0.4975 | 0.3133 | 0.3133 |
| $\nu_{13}^*$ [ ] | 0.264 | 0.1172 | 0.1809 | 0.3133 | 0.3133 |
| $\nu_{23}^*$ [ ] | 0.35 | 0.1172 | 0.2748 | 0.4707 | 0.4707 |
| $\gamma_\sigma^{-1} X$ [MPa] | 640.23 | 46.21 | 435.63 | 1285 | 772.7 |
| $\gamma_\sigma^{-1} X'$ [MPa] | -370.7 | -70.69 | -343.1 | -878.2 | -673.5 |
| $\gamma_\sigma^{-1} Y$ [MPa] | 38.1 | 46.21 | 76.44 | 38.1[b] | 38.1[b] |
| $\gamma_\sigma^{-1} Y'$ [MPa] | -82.18 | -70.69 | -174.7 | -82.18[b] | -82.18[b] |
| $\gamma_\sigma^{-1} S$ [MPa] | 30.17 | 124.5 | 85.06 | 30.17[b] | 30.17[b] |
| $\gamma_\sigma^{-1} R$ [MPa] | 18.97 | 18.97[b] | 18.97[b] | 18.97[b] | 18.97[b] |
| $\gamma_\sigma^{-1} T$ [MPa] | 6.21 | 6.21[b] | 6.21[b] | 6.21[b] | 6.21[b] |
| $m$ [ ] | 10 | 10 | 10 | 16.1 | 45.4 |
| $\rho$ [kg m$^{-3}$] | 1940 | 1940 | 1940 | 1600 | 1600 |

[a] Test result from a smaller fiber volume fraction. (Industry baseline pultrusion tests at 62% fiber volume fraction (Miller et al., 2019) )

[b] uniaxial glass composite value

**Table 2.** Mechanical properties of the isotropic constituents. All strength properties are design values (not the characteristic values) and are obtained with $\gamma_\sigma = 1.92$ (A larger $\gamma_\sigma$ is required by International Electrotechnical Commission (IEC) (IEC-61400-1, 2005) for the foam.)

| | $E$ [GPa] | $\nu$ | $\gamma_\sigma^{-1} X$ [MPa] | $\gamma_\sigma^{-1} X'$ [MPa] | $\gamma_\sigma^{-1} S$ [MPa] | $\rho$ [kg m$^{-3}$] |
|---|---|---|---|---|---|---|
| Gelcoat | 3.44 | 0.3 | - | - | - | 1235 |
| Foam | 0.1425 | 0.3194 | 2.083 | 1.563 | 1.250 | 130 |

assumed here to comprise $[(\pm 45°)(0°)_2]_s$ Saertex U14EU920-00940-T1300 and Saertex VU-90079-00830-01270—fabrics infused with Vantico TDT 177-155.



## 2.1 Elastic Properties

The only material found in a publicly available data set with fully characterized elastic properties was the uniaxial glass laminate. Other materials were largely limited to a few properties. The following three sections show how experimental values of the uniaxial glass laminate were distilled and summaries of how homogenization theory was used for the rest of the composites.

### 2.1.1 Experimentally Determined Properties

All nine of the elastic properties of the uniaxial glass laminate were obtained directly from experimental values (Samborsky

et al.). Since the tensile and compressive effective elastic moduli, $E^*$, were not equal, the arithmetic mean of the respective tensile and compressive value was utilized. For the shear moduli, $G^*$, the arithmetic mean of $G_{12}^*$ and $G_{21}^*$ was reported as $G_{12}^*$. Likewise, the arithmetic mean of $G_{13}^*$ and $G_{31}^*$ was reported as $G_{13}^*$ and the arithmetic mean of $G_{23}^*$ and $G_{32}^*$ was reported as $G_{23}^*$. The approach taken for the shear moduli is not applicable for the Poisson ratios, since in general, $\nu_{ij}^* \neq \nu_{ji}^*$, where $\nu_{ij}^*$ are Poisson ratios. $\nu_{12}^*$, $\nu_{13}^*$, and $\nu_{23}^*$ were used directly since $\nu_{21}^*$, $\nu_{31}^*$, and $\nu_{32}^*$ were obtained by loading the laminate in

unconventional ways.

### 2.1.2 Analytically Determined Properties

Homogenization, a formal subset of micromechanics, is routinely performed to enable efficient analysis and design. The biaxial and triaxial glass laminates both comprise uniaxial fabric placed at a number of angles that are then stitched together. Thus, they are both able to be respectively homogenized as a stack of homogeneous layers. For such a case, Yu (2012) has rigorously

proved that the in-plane strains and transverse stresses are uniform throughout the thickness of the laminate. Thus, the effective properties of the stack of layers can be obtained from a hybrid rule of mixtures in an exact manner. Thus, Yu's approach was adopted here for both of the laminates.

Note from Table 1 that one of the Poisson ratios of the biaxial glass is greater than 0.5. This is permissible since the commonly known maximum limit of 0.5 is exclusively for isotropic materials and this homogenized material is anisotropic. The repeating

unit cell considered for the biaxial glass composite was two layers of the uniaxial glass laminate with a stacking sequence of [45°/-45°]. The thickness fraction for each layer, defined as the layer thickness divided by the total thickness, was 0.5 for each layer. Note that actual thicknesses are not required for the analysis. As for the triaxial glass unit cell, it comprised three layers of the uniaxial glass laminate with a stacking sequence of [45°/0°/-45°], where the ±45° layers had a thickness fraction of 0.25 and the 0° layer had a 0.5 thickness fraction. All thickness fractions were derived from examining the SNL/MSU/DOE

Composite Materials for Wind Database (SNL/MSU/DOE).

### 2.1.3 Numerically Determined Properties

The layers of the carbon fiber laminates are all in the same direction and have not been characterized as fully as the uniaxial glass laminate. Thus, it is necessary to consider fiber-level details for the homogenization. The microstructure considered here was a hexangular packing of circular fibers, as shown in the RUC of Fig. 1. The local stress and strain fields in this





microstructure are not amenable to simplifications of 2.1.2. Thus, the effective elastic properties of the carbon fiber composites were obtained with a numerical model, known in micromechanics as representative volume element (RVE) analysis. The periodic boundary conditions were used because they are known to satisfy the well-known Hill-Mandel macrohomogeneity condition and only one unit cell is required in the RVE for converged properties. Further details of RVE analysis can be found in Yu (2019). The elastic modulus of the fiber along the fiber axis was calibrated from the corresponding tensile test data in

Miller et al. (2019) and the Voigt rule of mixtures. We obtained 230.26 GPa for the industry standard carbon fiber and 234.69 GPa for the heavy-tow fiber. The rest of the fiber properties were obtained from AS4 material properties in Herráez et al. (2020). As for the isotropic epoxy, the elastic modulus of 3.2 GPa and Poisson ratio of 0.347 was used.

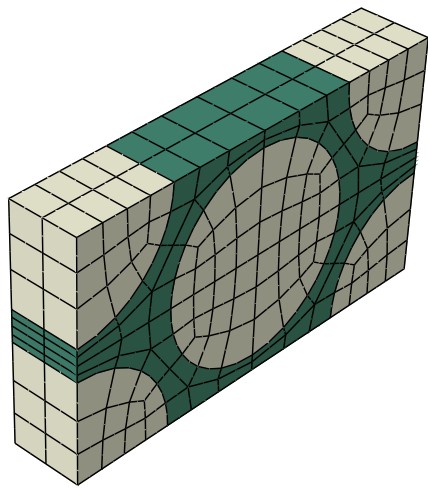

**Figure 1.** Representative volume element utilized to determine the effective elastic properties of the carbon fiber composites.

## 2.2 Rupture Properties

Table 1 also shows the strength values for each laminate. Note that $X$ is the longitudinal tensile strength, $Y$ is transverse (in-

plane) tensile strength, and their primed counterparts are the compressive strengths. $S$ is the in-plane shear strength, and $R$ and $T$ are the shear strengths associated with $\sigma_{23}$ and $\sigma_{13}$, respectively. Note that all of these are characteristic layer strengths.

All of the strengths for the uniaxial glass laminate were directly obtained from physical test data. For the remaining glass laminates, test data were only available for the in-plane strengths. The experimental data for the remaining laminates came from the SNL/MSU/DOE Composite Materials for Wind database (SNL/MSU/DOE). The out-of-plane shear strength properties, $R$

and $T$, for the biaxial and triaxial glass laminates, are set equal to the out-of-plane properties of the 0° laminate because they are assumed to be insensitive to fiber architecture and layup.

The SNL/MSU/DOE Composite Materials for Wind Database does not contain test data of the in-plane shear strength, $S$, for biaxial and triaxial glass laminates. These values were both determined with using a failure analysis technique that incorporated progressive damage analysis with classical lamination theory and LaRC03 (Davila and Camanho, 2003). Progressive damage





analysis was implemented as ply stiffness degradation after the ply was determined to have failed. The ultimate strength was determined by the last ply failure method.

$X$ and $X'$ for the uniaxial carbon laminates were obtained from the 95 % value (the same value used in Ennis et al. (2019)). The rest of the strength values for both carbon fiber laminates were assumed to be the same as the uniaxial glass laminate. The fatigue exponents, $m$, for the glass are standard values from DNVGL-ST-0376 (2015), whereas for the carbon fiber laminates, they are from Ennis et al. (2019).

## 3  Design Loads

Computational resources today cannot accommodate for fluid-structure interaction models in conjunction with a dynamic shell model of a blade for each design load case (DLC). Therefore, it is necessary to construct a reduced set of static load cases that is representative of the most critical loads from the beam model's time history analysis. The designs identified by the low-fidelity optimization were run through the aeroservoelastic solver OpenFAST (NREL, 2021a). Load equivalency between the OpenFAST beam models and the shell models constructed by NuMAD was established by equating the resultant forces and moments due to the stresses at various spanwise cross sections. Thus, the loading conditions were derived from the results of OpenFAST from NREL. The ultimate and fatigue limit states were evaluated from the International Electrotechnical Commission (IEC) standard (IEC-61400-1, 2005). Cases 1.1, 1.3, 1.4, 1.5, 5.1, 6.1, and 6.3 were considered for the ultimate limit states and 1.1 and 1.2 were considered for the fatigue limit states.

It is important to note that these cross-sectional resultants from OpenFAST are in coordinate systems that are in the deformed configuration and are aligned with the local principal axes (structural) of the cross section. Thus, those coordinate systems change with respect to span and time. Figure 2 contrasts the OpenFAST results coordinate basis vectors, $v_i^{(j)}$ ($j = 1, 2, 3, ..., k$), with those of the NuMAD loads system, $w_i$, and the ANSYS coordinate system, $x_i$, which are invariant. Note that the $x_i$ coordinate system is like the $w_i$ system but the first axis, $x_1$ points toward the leading edge instead of the flap direction.

A small angle between $v_3^{(j)}$ ($j = 1, 2, 3, ..., k$) and $w_3$ exists but is unaccounted for in NuMAD. Thus, it is assumed that $v_3^{(j)} = w_3 = x_3$ ($j = 1, 2, 3, ..., k$). The remaining angle between the $v_\alpha^{(j)}$ and $w_\alpha$ is referred to as $\mu$, which varies with span. Here and through the rest of this document, Greek indices assume 1 and 2 (except where explicitly indicated). These coordinates are illustrated in Fig. 3(a) and are related by

$$w_i = C_{iq} v_q^{(j)} \tag{1}$$

where $C_{ij}$ are the direction cosines that can be stored in a matrix as shown below

$$C = \begin{bmatrix} \cos(\mu) & \sin(\mu) & 0 \\ -\sin(\mu) & \cos(\mu) & 0 \\ 0 & 0 & 1 \end{bmatrix} \tag{2}$$

In this work, the principal axis angle was obtained by PreComp (Bir, 2006).





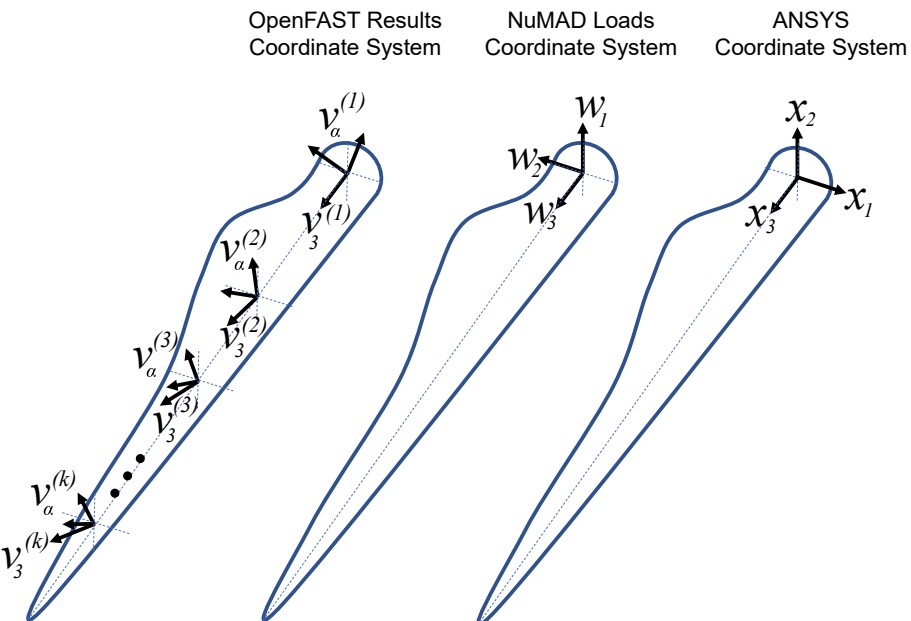

**Figure 2.** Comparison of the FAST results coordinate system with that of the NuMAD and ANSYS coordinate systems.

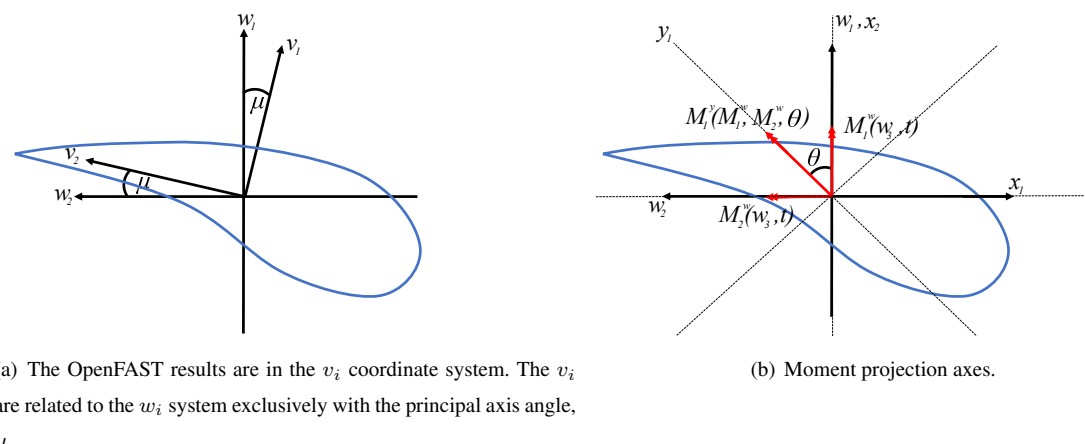

(a) The OpenFAST results are in the $v_i$ coordinate system. The $v_i$ are related to the $w_i$ system exclusively with the principal axis angle, $\mu$.

(b) Moment projection axes.

**Figure 3.** Coordinate systems utilized throughout.

Overall, two load categories were deemed necessary to properly analyze the most critical loads; those needed to evaluate the

maximum tip deflection and those needed for evaluating blade failure. Here, blade failure consists of ultimate failure, buckling, and fatigue failures. Both cases utilized the OpenFAST resultant axial forces, $F_3^w$, both bending moment components, $M_\alpha^w$, and the torsional moment, $M_3^w$. The superscript indicates the reference frame. These forces and moments will be referred to collectively with $P_i$ ($i = 1, 2, .., 4$), a generalized load vector, where $P_i = [F_3^w(t) \quad M_1^w(t) \quad M_2^w(t) \quad M_3^w(t)]^T$. Note that $P_i$



varies with span-wise location and that resultant shear forces that are transverse to the blade were assumed to be negligible for
establishing load equivalency for both load cases. For the tip-deflection case, the time at which the maximum tip deflection
from the beam model, $t^*$, was determined. Then the $P_i$, at a given cross section was defined by

$$P_i = \{F_3^w(t^*) \quad M_1^w(t^*) \quad M_2^w(t^*) \quad M_3^w(t^*)\}^T \tag{3}$$

The components were then transformed to the $x_i$ coordinate system.

IEC (IEC-61400-1, 2005) allows for lower factors of safety if numerous analysis directions are considered. Eight analysis
directions were considered here. Thus, the loads used to evaluate blade failure were obtained by letting $\theta$, as defined in Fig.
3(b), vary from $0°$ to $360°$ in load angle increments, $\Delta\theta$, of $45°$; yielding 8 FE load cases. The results for the load directions
$0° \leq \theta < 180°$ were obtained by

$$P_i = \begin{Bmatrix} \max\ (F_3^w(t)) \\ \max\ (M_1^y(t))\cos(\theta) \\ \max\ (M_1^y(t))\sin(\theta) \\ \max\ (M_3^w(t)) \end{Bmatrix} \qquad 0° \leq \theta < 180° \tag{4}$$

where

$$M_1^y = M_1^w(t, w_3)\cos(\theta) + M_2^w(t, w_3)\sin(\theta). \tag{5}$$

Load directions from $180° \leq \theta < 360°$ were, however, obtained by

$$P_i = \begin{Bmatrix} \max\ (F_3^w(t)) \\ \min\ (M_1^y(t))\cos(\theta) \\ \min\ (M_1^y(t))\sin(\theta) \\ \max\ (M_3^w(t)) \end{Bmatrix} \qquad 0° \leq \theta < 180° \tag{6}$$

Note in Eq. (6) that the minimum of $M_1^y$ is found instead of the maximum but $\theta$ still ranges from $0°$ to $180°$. Unlike the
resultants used for the deflection analysis, which all occurred at a specific time in the OpenFAST simulations, each of the axial
force, torsion, and bending moment resultants along the span could possibly come from different times. Thus, it is an artificial
distribution suitable for design use. Also, unlike the deflection case, the two bending-moment components (i.e., flapwise and
edgewise bending) at a span location were projected onto eight directions, as defined by $y_1$ in Fig. 3(b). The loads in these
eight directions were also transformed to the $x_i$ system.

For any given distribution of bending moments on the blade, whether for the deflection analysis or evaluating failure, the
forces to be applied to the blade were obtained by writing mechanical equilibrium expressions for each spanwise position of
interest. Fig. 4 shows the known spanwise bending moments, $M_i$ acting at a distance, $z_i$, from the blade root and the forces
to be solved, and $F_i$ acting at $\bar{z}_i$, where $\bar{z}_i = \frac{1}{2}(z_i + z_{i+1})$. The forces to be applied are found by solving the following linear





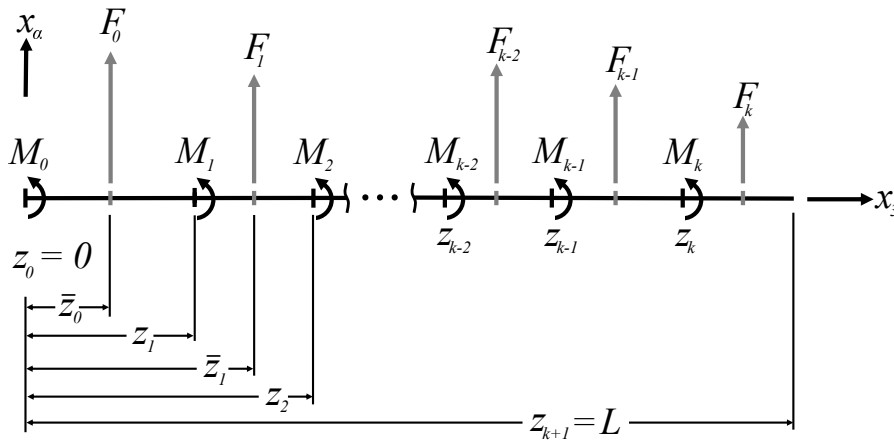

**Figure 4.** Freebody diagram used to determine transverse loads from a given distribution of moments along the blade span.

system of equations for $F_i$

$$M_i = \sum_{j=i}^{k} F_j(\bar{z}_j - z_i) \tag{7}$$

where $k$ is the number of cross sections with resultant moment data. Because OpenFAST only outputs this moment data at a
maximum of 10 blade cross sections, the moments were with a shape-preserving piecewise cubic interpolation. In this way,
force distributions parallel to $w_1$ and $w_2$ are obtained. Finally, these force distributions were transformed to nodal forces on the
blade outer shell by following Berg et al. (2011) but with an extension to apply axial forces and twisting moments.

## 4   Main Optimization Loop

Once the preprocessing steps were completed for each design, the main structural optimization process was performed in
NuMAD. The following sections describe the definition of the optimization objective function, design variables, and algo-
rithms/methods used.

   The overall objective of the high-fidelity structural optimization in NuMAD was to minimize mass of the blade structure
subject to the following four performance constraints: 1) resistance against material rupture, 2) resistance against fatigue
damage over operating life span of the turbine, 3) resistance against structural buckling, 4) maximum tip deflection not to
exceed limit for tower clearance, and 5) natural vibration frequencies in the flap and edge directions not to coincide with
the main rotor harmonics. Minimization of mass is ideal for nearly any mechanical device, if only for the sake of reducing
consumption and cost of materials for the component itself. For the present application, there is additional importance from a
system-level perspective



## 4.1 Constraints

For each iteration in the optimization, NuMAD calls ANSYS to obtain the constraint values and stress data, as well as to compute blade mass for the objective function. Figure 5 illustrates the FE workflow for a single iteration of the optimization. The total blade mass and frequencies are first computed. Then the load data that is associated to the maximum tip deflection from OpenFAST was applied to the blade for a single linear-static analysis to obtain the tip deflection. Then blade failure (comprising material rupture and buckling) was evaluated by considering each load direction, $\theta$. For a given load angle, a material rupture index was evaluated at every location and layer of the blade from a linear-static analysis. This was computed by utilizing a Tsai-Wu failure criteria that neglects any effects due to transverse normal stress (ANSYS, 2017). The maximum rupture index for the whole blade is stored, $\sigma_\theta$ $(i = 1, 2, 3, ...n_\theta)$, as well as stress data for each layer of every element for the fatigue postprocessor described in Sect. 4.1.1. Note that $n_\theta$ is the number analysis directions and that the default value of -1 in ANSYS was used for the three shear coupling coefficients in the Tsai-Wu criteria. The buckling load factor associated to the lowest mode, $\beta_\theta$ $(i = 1, 2, 3, ...n_\theta)$, was then stored from a subsequent eigenvalue buckling analysis of the entire blade. This process repeats for every load direction. It is notable that geometric and material nonlinearity is unaccounted for. These constraints are summarized mathematically, as follows:

$$\text{Material Rupture:} \qquad \sigma_\theta < \frac{1}{\gamma_\sigma} \qquad \Rightarrow \qquad \sigma_\theta \gamma_\sigma < 1 \tag{8}$$

$$\text{Fatigue Damage:} \qquad D < \frac{1}{\gamma_D} \qquad \Rightarrow \qquad D\gamma_D < 1 \tag{9}$$

$$\text{Buckling:} \qquad \beta_\theta > \gamma_\beta \qquad \Rightarrow \qquad \frac{\gamma_\beta}{\beta_\theta} < 1 \tag{10}$$

$$\text{Maximum Tip Deflection:} \qquad \gamma_u u_{tip} < u_{max} \qquad \Rightarrow \qquad \frac{\gamma_u u_{tip}}{u_{max}} < 1 \tag{11}$$

$$\text{Minimum Flap Frequency:} \qquad f_{flap} > \gamma_f(3f_{rot}) \qquad \Rightarrow \qquad \frac{\gamma_f(3f_{rot})}{f_{flap}} < 1 \tag{12}$$

$$\text{Minimum Edge Frequency:} \qquad f_{edge} > \gamma_f(4f_{rot}) \qquad \Rightarrow \qquad \frac{\gamma_f(4f_{rot})}{f_{edge}} < 1 \tag{13}$$

$$\text{Flap/Edge Frequency Separation:} \qquad f_{edge} > \gamma_f f_{flap} \qquad \Rightarrow \qquad \frac{\gamma_f f_{flap}}{f_{edge}} < 1 \tag{14}$$

In the above, $u_{tip}$ is the highest tip deflection of the blade in the direction normal to the rotor's sweep plane, and $u_{max}$ is the highest allowable blade deflection to ensure tower clearance. $f_{flap}$ and $f_{edge}$ are the first natural vibration frequencies in the flapwise and edgewise directions, respectively, and $f_{rot}$ is the frequency of the rotor at the rated speed. $\gamma_D$, $\gamma_\beta$, $\gamma_u$, and $\gamma_f$ are the factors of safety applied to fatigue, buckling, tip deflection, and natural frequencies, respectively ($\gamma_\sigma$ defined in Sect. 2). Note that $\gamma_D$ is set to 1.0 because the fatigue factor of safety is implemented in Eq. (18). The rest of the design values are listed in Table 3.

### 4.1.1 Fatigue Damage Postprocessor

NuMAD 3.0 now includes the capability to evaluate fatigue damage for every material layer at a discreet number of spanwise stations. First, stress data in the local layer coordinate system for every element and every layer is read into MATLAB® , as




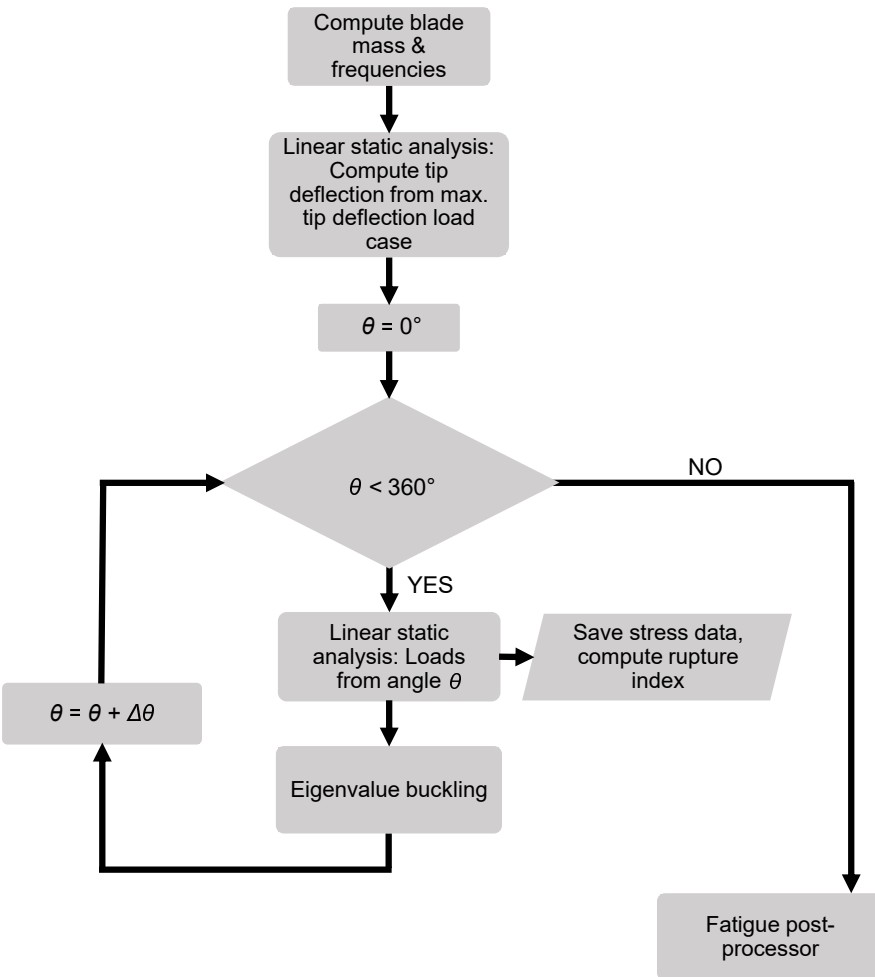

**Figure 5.** FE operations workflow in a single iteration of the optimization.

**Table 3.** Utilized factors of safety.

| Material Rupture, $\gamma_\sigma$ | Buckling $\gamma_\beta$ | Deflection $\gamma_u$ | Natural Frequencies, $\gamma_f$ |
|---|---|---|---|
| 1.74 | 1.58 | 1.42 | 1.1 |

well as element location along $x_3$. This is done for both the $y_1(\theta)$ and $y_1(\theta + 90°)$ directions to consider the additive fatigue damage brought by two orthogonal load directions. The script then loops through each blade cross section (a total of $n_{sec}$ cross sections), comprising the root and each OpenFAST gage location. Because a time-history analysis in the shell model was avoided, at each of these locations Markov matrices for both orthogonal directions are constructed from rainflow cycle counting



on the $M_1^y(M_1^v, M_2^v, \theta)$ and $M_1^y(M_1^v, M_2^v, \theta + 90°)$ time histories (see Fig. 4). Thus, the NREL Crunch (Buhl, 2003) program

was setup to analyze each wind speed file with "calculated channels" for $M_1^y$ at each spanwise location and direction. Because the time-history data from OpenFAST comprises 10 min of simulated time, the cycle counts are extrapolated to account for the design life of 30 yr by considering a Rayleigh distribution of the annual wind speed. Fatigue damage is then calculated for all layers in elements within 0.75 m of each cross section.

In a given element and a given layer, fatigue damage is calculated by converting the Markov matrix from being defined

with moment amplitudes and means to the amplitudes and means of the predominant normal stress component with pseudo-constitutive relations

$$\sigma_a = \alpha M_a \qquad \sigma_m = \alpha M_m \qquad (15)$$

where $\sigma_a$ and $\sigma_m$ are predominant normal stress amplitudes and means, respectively, and $M_a$ and $M_m$ are the amplitude moments and mean moments, respectively. The scalar parameter $\alpha$ is obtained by

$\alpha = \sigma_{11}/M_r(\theta, x_3)$ (16)

where $\sigma_{11}$ is the normal stress in the fiber direction of a given layer, and $M_r(\theta, x_3)$ is the magnitude of the resultant moment applied to the cross section. Thus, the moment-based Markov matrices are converted to stress-based Markov matrices suitable for damage calculations. The well-known Palmgren-Miner (1945) linear damage rule is then utilized to find the total damage in a layer due to cycles along the $y_1(\theta)$ direction as

$d^\theta = \sum_{i=1}^{n_a} \sum_{j=1}^{n_m} \frac{\kappa(\sigma_a^i, \sigma_m^j)}{N_f}$ (17)

where $n_a$, and $n_m$ are the number of bins for the stress amplitude and mean stress, respectively, $\kappa$ are the cycle counts, and $\sigma_a^i$ and $\sigma_m^j$ are the stress amplitudes and mean stresses for the layer, respectively. $N_f$ is the number of cycles to failure and were computed with a shifted Goodman approach as

$$N_f = \left( \frac{X + |X'| - |2\sigma_m^j \gamma_\sigma - X + |X'||}{2\sigma_a^i \gamma} \right)^m \qquad (18)$$

from the DNVGL standard (DNVGL-ST-0376, 2015). $\gamma$ is the fatigue factor of safety and was taken to be equal to $\gamma_\sigma$ because more than two load directions were being analyzed but less than 12.

Everything in the preceding paragraph was also done for the $y_1(\theta + 90°)$ direction in parallel. The fatigue damage of the layer then becomes

$$d = d^\theta + d^{\theta+90°} \qquad (19)$$

where $d$ is the fatigue damage of the layer due to cycles along the orthogonal $y_1(\theta)$ and $y_1(\theta + 90°)$ directions. A fatigue damage fraction is assigned to the $i^{\text{th}}$ cross section by taking the maximum damage of every element within 0.75 m of the cross section and the maximum of each layer in those elements as in

$$D_{\theta,i} = \max\ (d) \qquad (\theta = 1, 2, 3, ... n_\theta/2) \qquad (20)$$





The process repeats for the remaining cross sections. Then, the current direction is incremented by $45°$ and the process repeats for $n_\theta/2$ times.

## 4.2 Objective Function

In defining the objective, or fitness function, it was important to consider that these constraints are all nonlinear and not expressible in closed form with respect to design parameters. Also, because gradient-based optimization was employed in the design process, it was advantageous to define an objective function that is as smooth and continuous in the design space as possible. A constraint aggregation approach has often been used to apply stress constraints to structures in gradient-based optimization while preserving smoothness in the objective (Kreisselmeier and Steinhauser, 1980). With these things in mind, the objective function was defined with the performance constraints enforced as penalty terms, as follows:

$$L = m_b + c_1 \begin{pmatrix} \frac{1}{n_\theta}\sum_{\theta=1}^{n_\theta}\left\{\frac{1}{n_{el}}\sum_{i=1}^{n_{el}}(\sigma_{\theta,i}\gamma_\sigma)^{c2} + \frac{1}{n_{modes}}\sum_{i=1}^{n_{modes}}\left(\frac{\gamma_\beta}{\beta_{\theta,i}}\right)^{c2}\right\} \\ +\frac{2}{n_\theta}\frac{1}{n_{comp}}\frac{1}{n_{sec}}\sum_{\theta=1}^{\frac{n_\theta}{2}}\sum_{i=1}^{n_{sec}}\sum_{j=1}^{n_{comp}}(D_{\theta,ij}\gamma_D)^{c2} \\ +\left(\frac{u_{tip}}{u_{max}}\right)^{c2} + \left(\frac{\gamma_f(3f_{rot})}{f_{flap}}\right)^{c2} + \left(\frac{\gamma_f(4f_{rot})}{f_{edge}}\right)^{c2} + \left(\frac{\gamma_f f_{flap}}{f_{edge}}\right)^{c2} \end{pmatrix} \tag{21}$$

For material rupture and buckling, the key results are summed, or, in other words, aggregated over the number of elements, $n_{el}$, and the number of buckling modes analyzed, $n_{modes}$, respectively, over $n_\theta$ transverse loading directions, as explained in Sect. 3. For fatigue damage, the results are summed over the number of blade cross sections analyzed for fatigue damage, $n_{sec}$, the number of blade components, $n_{comp}$, and over one half the number of loading directions, as the loading directions are analyzed in pairs for fatigue analysis. $D_{\theta,ij}$ represents the maximum damage fraction in cross section $i$ and blade component $j$, under load direction $\theta$. The constants $c_1$ and $c_2$ are case-dependent, and ideally chosen so that the constraint aggregation terms are negligible for a design that fully satisfies all constraints but become dominant when any constraint is violated. Values of $c_1$ = 60000 and $c_2 = 2$ were found to be suitable, and $c_1$ was scaled to match the approximate mass of the blade so that the penalty terms have an effective balance with mass when the constraints become active. $c_2$ is an exponent in the penalty term, which has a much stronger impact on rate of growth and smoothness. Finally, at any given design state in the optimization process, all the terms in the objective function are evaluated by running FE analyses on the blade under the loads derived from OpenFAST using ANSYS (see Sect. 3 for more details).

## 4.3 Design Variables

The structural optimization in NuMAD was performed by modifying geometric aspects of each blade's design with regard to material thicknesses. Each blade comprises the following main components: spar caps, leading edge panels, and trailing edge panels (each on both the pressure side and the suction side of the airfoil cross section), leading edge reinforcement, trailing edge reinforcement, and two shear webs connecting the suction side and pressure side spar caps. There is also a layer of skin on the outer and inner surfaces along the entire spanwise length of the blade. These main components are illustrated in a cross section view in Fig. 6.





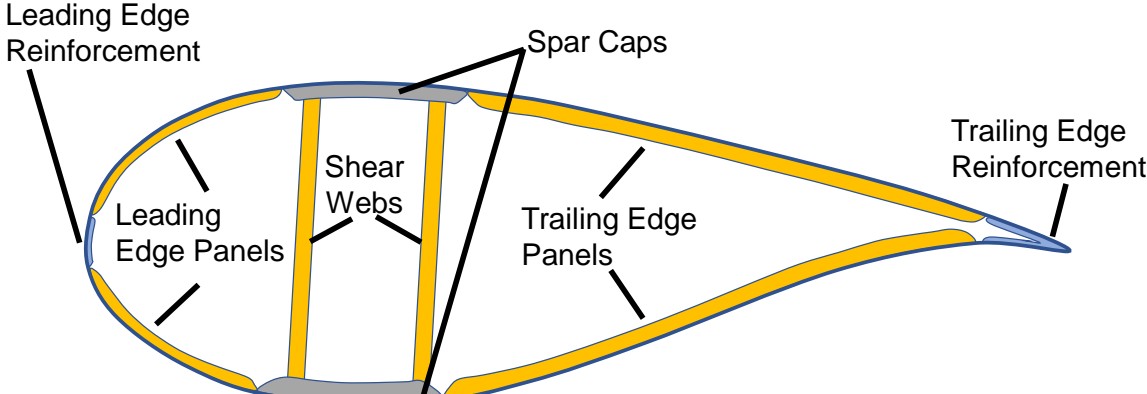

**Figure 6.** Cross-sectional component breakdown of a wind turbine blade.

The data structure that NuMAD uses to define a blade includes geometric specifications such as width and thickness for each of these components along the span of the blade. The design variables defined for structural optimization in the present study

control the spanwise thicknesses of all the components, the width of the spar caps (defined as a single nominal value for the entire length), and the spanwise extent of the shear webs. Note that each component often comprised different materials. For example, the trailing edge panels comprised a triaxial glass/foam/triaxial glass layup. The thickness of each layer was allowed to vary individually. Every laminate orientation was fixed such that $E_1^*$ "pointed toward" the blade tip and remained tangential to the shell reference surface.

The thickness of each component along the span of each blade was defined at 30 equally spaced points from the root to the tip. Six key points along the length were defined as design variables for each component, with the intermediate points interpolated as a smooth spline to define the spanwise distribution.

### 4.4 Algorithms and Other Methodology

As described in Sect. 1, the initial blade designs were received from collaborators at NREL, following the low-fidelity opti-

mization (Bortolotti et al., 2021). Initially, the designs generally did not satisfy the structural performance constraints under the loads determined by the OpenFAST aeroelastic analysis. The high-fidelity structural optimization updated the design to a state in which the performance constraints were satisfied while simultaneously minimizing the mass of the blade. The optimization was performed using the MATLAB built-in function, *fmincon*, a general nonlinear gradient-based optimization solver. The interior point variation was used, obtaining the gradient/sensitivities of the objective with central differencing.

While the initial designs from the NREL Wind-Plant Integrated System Design & Engineering Model (WISDEM® ) tool (NREL, 2021b) did not meet all constraints, they were carefully developed based on experience and previous tried-and-true designs. In the interest of staying within reasonably manufacturable parameters and preserving a smooth iteration cycle between high-fidelity and low-fidelity optimization, it was desirable to find an optimized structural design as close to the initial state as



possible while still achieving the objectives. Gradient-based optimization adjusts the design variables that most strongly affect
the objective, moving to a more favorable state with minimal overall change in design. Once the main automated phase of
optimization was completed, some final postprocessing steps were taken, as described in the next section.

### 4.5 Optimization Postprocessing

After the automated structural optimization was completed for each blade design, the results were inspected visually for quality
and verification using the ANSYS results viewing window. If needed, minor manual adjustments were made to the design
where there could potentially be problems with fabrication or other issues. The model output was examined for factors such as
percentage of failed elements and exceeded buckling load factors to evaluate whether any key results were driven by poor mesh
quality or fictitious physical effects. The updated design was then sent back to collaborators at NREL for further iteration. If
applicable, the new design was evaluated for transportation by rail (BAR-DRG, BAR-DRC, and BAR-URC), and modifications
were made as necessary. The global optimization cycle was repeated, beginning with system-level design and optimization until
converging on a final design.

## 5 Results

### 5.1 Verification

It was first necessary to verify that the shell model coincided with the beam model at NREL. The blade stiffness and total blade
mass between each model were verified during one of the design iterations. An identical yaml file was used for the stiffness
and mass verification, respectively.

Because BeamDyn (NREL, 2021c) is the highest-fidelity beam model from the NREL tools and because loads from Beam-
Dyn inform both low-fidelity and high-fidelity optimizers, the deflections between BeamDyn and ANSYS were compared in
lieu of direct stiffness comparisons. Since both models make use of a reference line to build the geometry, it was convenient
to track its deflection. The displacement of the reference line is easily obtained from the NREL beam analysis because it is
part of the formulation. The displacement of the reference line is not easily obtained from a shell (or solid model) since a
reference line is not an intrinsic part of the governing differential equations. The resulting displacement field from a shell (or
solid) model is a combination of cross-sectional translations, rotations, and warping; an unknown combination at that. Merely
using the arithmetic average of displacements from the nodes at a cross section is an approximate approach. Nonetheless, this
approach was taken here and thus an approximate comparison to BeamDyn was made. No special effort was made to remove
cross-sectional rotations and warping effects from the shell model. Figure 7 shows the flapwise and edgewise deflection due
to a uniform flapwise line load; labeled as $q_2$ (acts parallel to $x_2$). It was applied along the length of the reference axis during
a design iteration for BAR-UAG. Figure 7(a) shows a good comparison for the flapwise deflection. The comparison of the
edgewise deflections in Fig. 7(b) is not as good as the flapwise deflections but satisfactory nonetheless. Note that the arithmetic
mean displacement of a cross section, $\bar{u}_i$, acts parallel to $x_i$ (see Fig. 2).





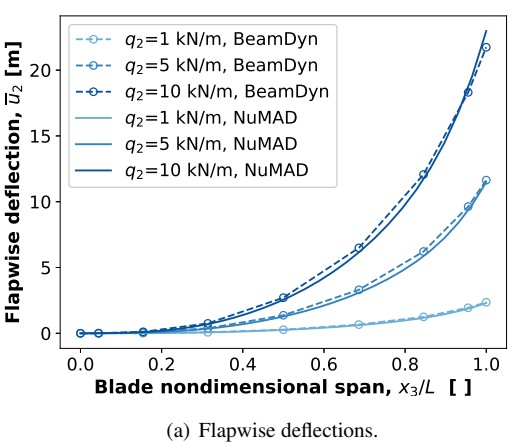

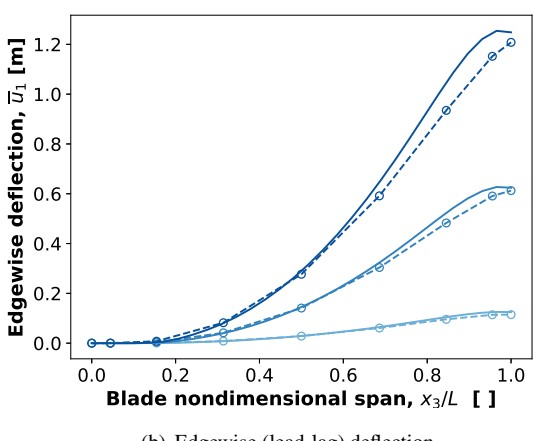

| (a) Flapwise deflections. | (b) Edgewise (lead-lag) deflection. |

**Figure 7.** Flapwise and edgewise static deflection comparison between NuMAD and BeamDyn during a design iteration for BAR-UAG.

**Table 4.** Blade mass comparison between WISDEM and NuMAD during a design iteration for BAR-UAG.

| Laminate | WISDEM [kg] (NREL beam model) | NuMAD [kg] (SNL shell model) | Difference [%] |
|---|---|---|---|
| Gelcoat | 529 | 524 | -0.8 |
| Glass Uni. | 30,159 | 32,207 | 6.4 |
| Glass Biax. | 1,477 | 1,378 | -7.2 |
| Glass Triax. | 31,790 | 33,483 | 5.1 |
| Foam | 3,409 | 3,035 | -12.3 |
| Total | 67,363 | 70,628 | 4.6 |

During the course of the project, the blade mass between WISDEM and NuMAD for BAR-UAG was also verified. Table 4 shows the total mass of each material as well as the total mass of the blade. The codes are within 5 % of each other. The discrepancies were traced primarily to modeling inaccuracies of complex geometries and layup information.

## 5.2 Optimized Blade Designs

The results of the blade design optimization provide insight into understanding the advantages, disadvantages, and potential
challenges in different approaches for low-specific-power wind turbine design. The final design states following high-fidelity optimization for all the blade models, with regard to mass and state of performance constraints, are summarized in Table 5. In general, it is shown that the total mass of a blade can be reduced by adopting a flexible design and alleviating the constraint on tip deflection, as for designs BAR-DRG, BAR-DRC, BAR-DRCHT, BAR-URC, and BAR-URCHT. These designs then





**Table 5.** Final mass and state of design constraints for optimized blade designs. Constraint index (unitless) values expressed as defined in Eqs. (8)–(13), with limiting constraints denoted with bold font.

| Design | Mass [kg] | Rupture | Fatigue | Buckling | Deflection | Flap Freq. | Edge Freq. |
|---|---|---|---|---|---|---|---|
| BAR-UAG | 67,073 | 0.74 | 0.084 | 0.73 | **0.99** | **0.99** | 0.74 |
| BAR-DRG | 55,201 | **0.99** | 0.60 | **0.99** | – | **0.99** | **0.99** |
| BAR-DRC | 43,807 | **0.98** | 0.013 | **0.98** | – | 0.90 | 0.75 |
| BAR-USC | 50,765 | 0.69 | 0.023 | 0.71 | **0.99** | 0.83 | 0.80 |
| BAR-URC | 43,187 | **0.99** | 0.0041 | **0.98** | **0.96** | 0.90 | 0.74 |
| BAR-DRCHT | 44,061 | **0.98** | 0.012 | **0.98** | – | 0.90 | 0.76 |
| BAR-USCHT | 50,539 | 0.70 | 0.023 | 0.75 | **0.99** | 0.81 | 0.82 |
| BAR-URCHT | 44,615 | **0.99** | 0.0030 | **1.0** | **0.96** | 0.90 | 0.74 |

became driven mainly by material rupture and buckling constraints. In contrast, the upwind designs—BAR-UAG, BAR-USC,
and BAR-USCHT—took on a higher final mass and were mainly driven by the constraint of tip deflection.

However, the flexible models were seen to generally have a more limited range of feasible design space. For BAR-DRG, for example, both the maximum section stiffness allowable for rail transportability (Bortolotti et al., 2021) and the minimum flap frequency constraint are near their prescribed limits. Because these two constraints are inherently competing, there exists only a small window of feasibility in the spar cap design that satisfies both under the assumptions used.

BAR-DRC has an advantage over BAR-DRG in that the higher stiffness-to-mass ratio of the carbon fiber design brings the flap and edge frequencies up well into the safe range. There are still the material rupture and buckling constraints, however, they remain in opposition of the rail-transportability constraint. Still, if a flexible design is to be pursued, the evidence would support the adoption of a carbon fiber composite spar cap.

None of the models had fatigue damage as a driving factor in the optimization. The nearest model being affected is BAR-
DRG, at a maximum damage factor of 0.597. Though on the surface this could be interpreted as a positive result, there may be reason to investigate the analysis approach and further verify it in future work. It is widely known in industry, for example, that when a blade failure arises from fatigue damage, it nearly always nucleates from vulnerable areas of load concentration, such as bolts and joints in panels. These features drive the source and propagation of fatigue damage, yet they are not realistically captured in the shell models currently employed. Even if such features were modeled more realistically, fatigue life inherently
has a high degree of uncertainty, and it may turn out that setting/verifying the appropriate safety factors to use for modeling applications like this could be the most productive course of action.

Table 6 shows the design load cases, as listed in Sect. 3, which dominated the maximum loads experienced by the blade in the aeroelastic simulations and used in the structural analysis. Although the load distribution applied to the FE model is derived from maximum forces and moments taken throughout the length of the blade and throughout all simulations as previously
explained, Table 6 highlights the source of the maximum bending moment at the root in the flap and edgewise directions, as well as the maximum tip deflection. It is evident that the dominant loads generally come from certain case simulations, in





**Table 6.** Driving design load cases and limiting components for optimized blade designs. Included is the margin by which the indicated design load case exceeds the others.

| Design | DLC of Max Flap Moment, (Margin of Exceedance) | DLC of Max Edge Moment, (Margin of Exceedance) | DLC of Max Deflection, (Margin of Exceedance) | Limiting Component(s) |
|---|---|---|---|---|
| BAR-UAG | 1.3 (8.4%) | 1.4 (11.0 %) | 1.4 (0.7 %) | Spar caps |
| BAR-DRG | 1.4 (79.8 %) | 1.4 (51.4 %) | 1.4 (66.9 %) | Shear web skins, leading edge suction-side panels |
| BAR-DRC | 1.4 (112.6 %) | 1.4 (123.8 %) | 1.4 (99.1 %) | Shear web skins, trailing edge reinforcement, suction-side spar cap |
| BAR-USC | 1.3 (12.7 %) | 1.4 (23.8 %) | 1.3 (6.9 %) | Spar caps |
| BAR-URC | 1.3 (8.3 %) | 1.4 (23.4 %) | 1.3 (3.6 %) | Shear web skins, trailing edge reinforcement, suction-side spar cap |
| BAR-DRCHT | 1.4 (112.6 %) | 1.4 (123.8 %) | 1.4 (99.1 %) | Shear web skins, trailing edge reinforcement, suction-side spar cap |
| BAR-USCHT | 1.3 (12.7 %) | 1.4 (23.8 %) | 1.3 (6.9 %) | Spar caps |
| BAR-URCHT | 1.3 (8.3 %) | 1.4 (23.4 %) | 1.3 (3.6 %) | Shear web skins, trailing edge reinforcement, suction-side spar cap |

particular 1.4, with some from 1.3 from the IEC standard design load cases. This trend is not universal but is consistent among the examined designs.

Regarding the individual designs, a clear observation is that, for the downwind designs (BAR-DRG, BAR-DRC and BAR-DRCHT), loads are distinctly dominated by design load case 1.4, whereas the upwind design loads are more balanced between simulation cases. Although DLC 1.4 is prominent for all designs, when it dominates in upwind designs, it does so by a notably smaller margin compared to downwind designs. The implication is that downwind designs will generally be more specifically driven by sudden, abrupt events like gusts, whereas upwind designs would be somewhat more dictated by normal operation. Furthermore, the dominance of DLC 1.4 in downwind designs may suggest that if some measures were taken to alleviate loads in sudden gusts, such as active aero devices or bend-twist coupling, it could perhaps allow the blade mass to come down further yet, or increase robustness of the design.

Table 6 also indicates any components of the blade that are limiting with regard to critical driving constraints. For blades driven by material rupture, the shear web skin and trailing edge reinforcement commonly showed peak failure indexes, particularly near the root of the blade where the shear webs begin and near-relatively sudden changes in the cross section. Figures 8, 9, and 10 show the Tsai-Wu failure index distribution for the critical loading for BAR-DRG, BAR-DRC, and BAR-URC, respectively, as these were the blades most governed by material rupture. These same designs also showed buckling as a driving



constraint, mainly in leading edge panels and spar caps, about 20 m from the root of the blade. Figures 11, 12, and 13 show the buckling modes nearest to the critical load factor for BAR-DRG, BAR-DRC, and BAR-URC, respectively.

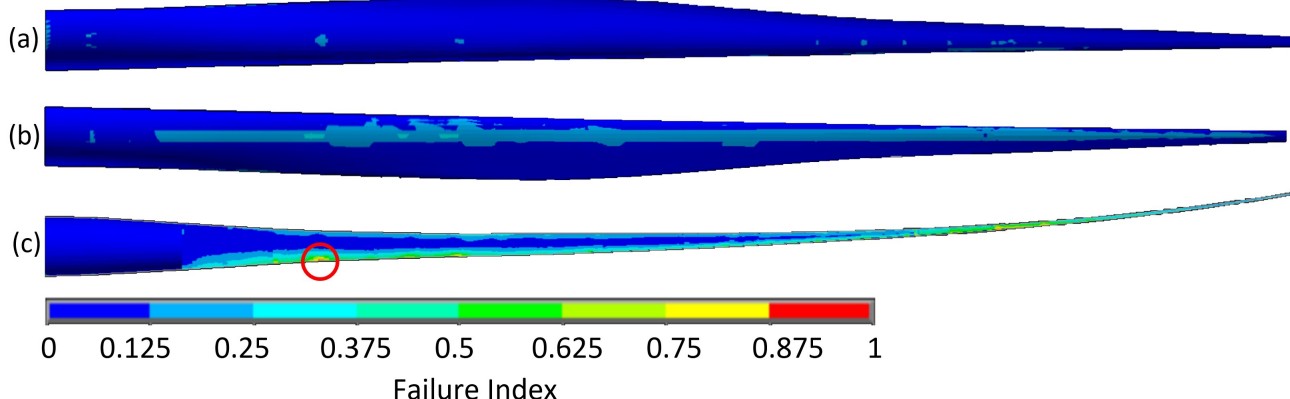

**Figure 8.** Tsai-Wu failure index for the optimized BAR-DRG design under critical loading, shown from the (a) pressure side, (b) suction side, and (c) cut-away view, showing the leading shear web. Vulnerable areas circled in red.

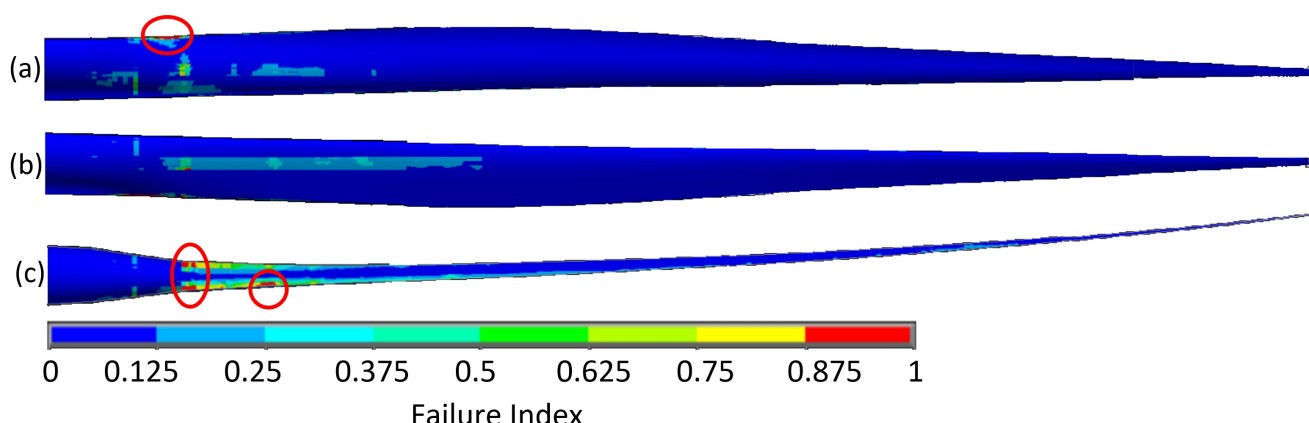

**Figure 9.** Tsai-Wu failure index for the optimized BAR-DRC design under critical loading, shown from the (a) pressure side, (b) suction side, and (c) cut-away view, showing the leading shear web. Vulnerable areas circled in red.

In comparison of the optimized design for blades featuring carbon fiber composite spar caps using industry-standard unidi-
rectional carbon fiber composite vs. heavy-tow carbon fiber composite; only minor differences were seen in the final designs between the two materials. A basic trend was seen that for designs with the heavy-tow composite spar caps, which have a slightly lower ultimate strength, the spar cap thickness generally increased marginally compared to the equivalent designs us-




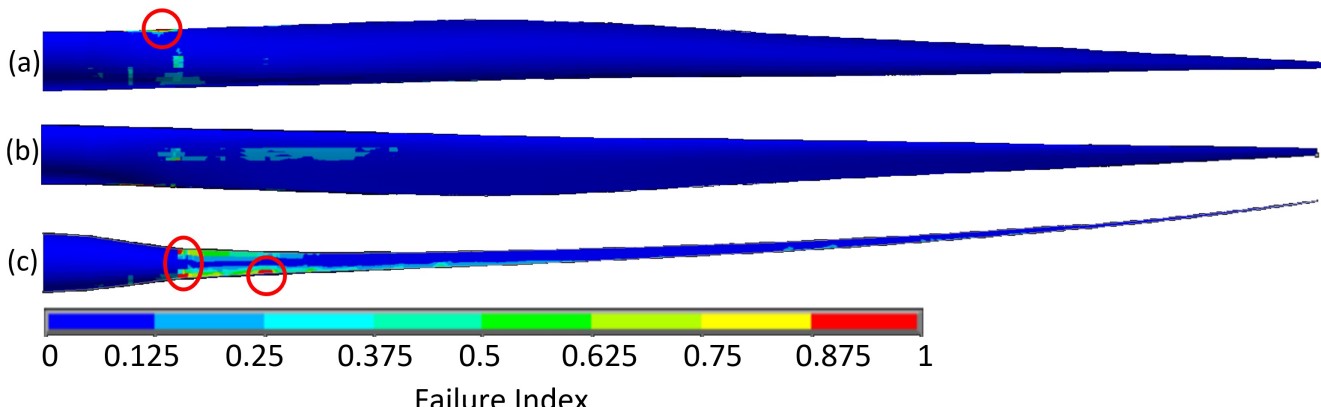

**Figure 10.** Tsai-Wu failure index for the optimized BAR-URC design under critical loading, shown from the (a) pressure side, (b) suction side, and (c) cut-away view, showing the leading shear web. Vulnerable areas circled in red.

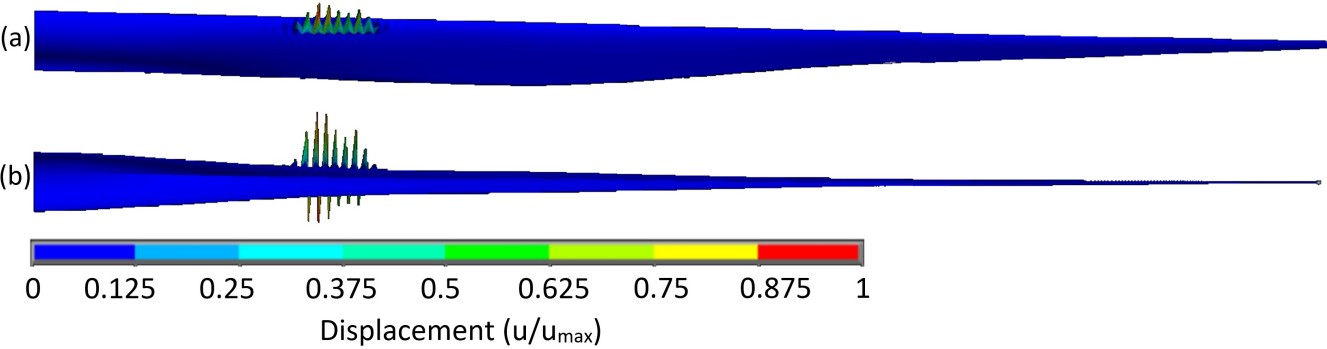

**Figure 11.** Displacement plot of the most critical buckling mode for the optimized BAR-DRG design viewed from the (a) suction side and (b) trailing edge.

ing the industry-standard carbon fiber composite. Figures 14, 15, and 16 show the final optimized spar cap thickness profile for BAR-DRC/BAR-DRCHT, BAR-USC/BAR-USCHT, and BAR-URC/BAR-URCHT, respectively.

A detailed cost analysis was performed for each blade design by collaborators at NREL (Bortolotti et al., 2021). The results for blades featuring carbon fiber composite spar caps are summarized in Table 7. Due to the significantly reduced cost of heavy-tow carbon fiber composite compared to the industry-standard composite, combined with the slightness of change in optimized designs, the analysis indicates roughly a 10 % cost savings in blade manufacture, which is made possible by employing the heavy-tow composite instead of the industry standard. It would seem to be worth investigating this avenue further to verify if

the findings are correct, and what additional factors could affect the final cost that may not be accounted for in this study.





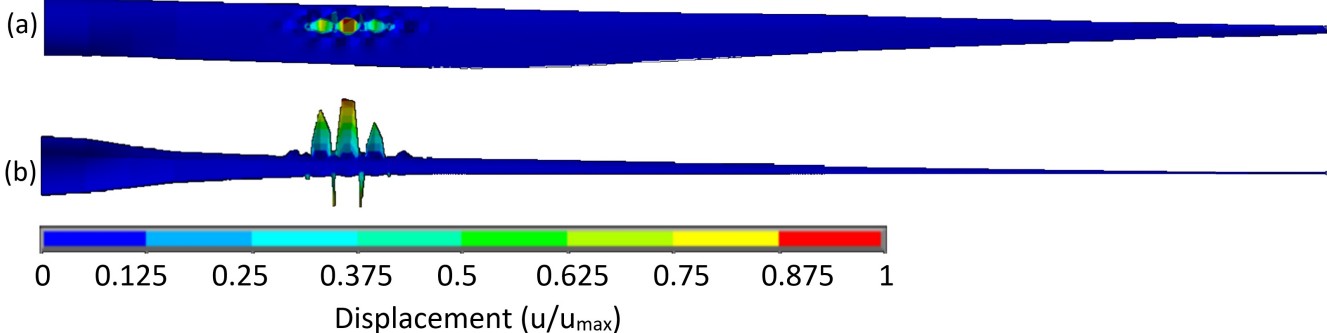

**Figure 12.** Displacement plot of most critical buckling mode for optimized BAR-DRC design viewed from (a) suction side, and (b) trailing edge.

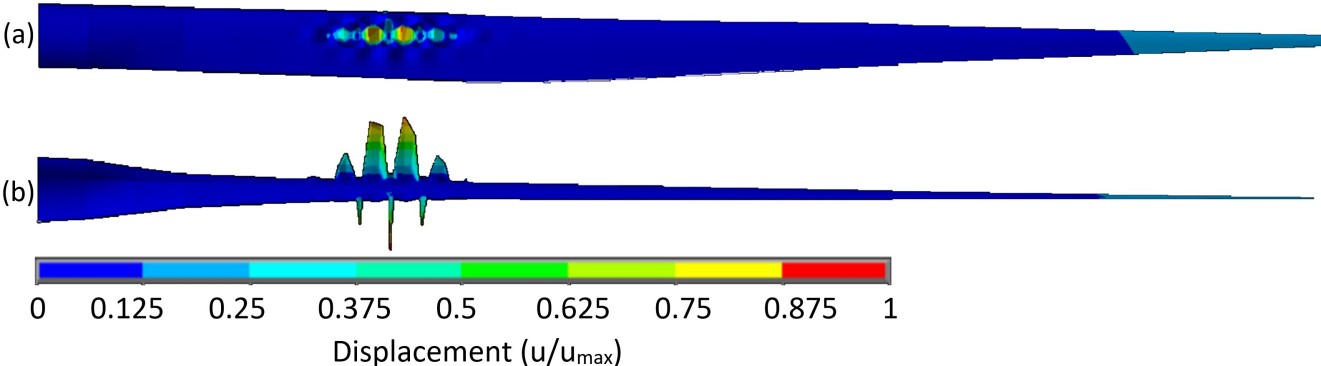

**Figure 13.** Displacement plot of most critical buckling mode for optimized BAR-URC design viewed from (a) suction side, and (b) trailing edge.

**Table 7.** Cost comparison of final blade designs using industry-standard carbon fiber composite spar caps vs. heavy-tow carbon fiber composite spar caps.

| Design | Total Cost, Industry Standard | Total Cost, Heavy-Tow | Percent Reduction |
|---|---|---|---|
| BAR-DRC | $471,827 | $412,610 | 12.6 % |
| BAR-USC | $563,470 | $512,990 | 8.9 % |
| BAR-URC | $465,864 | $416,426 | 10.6 % |



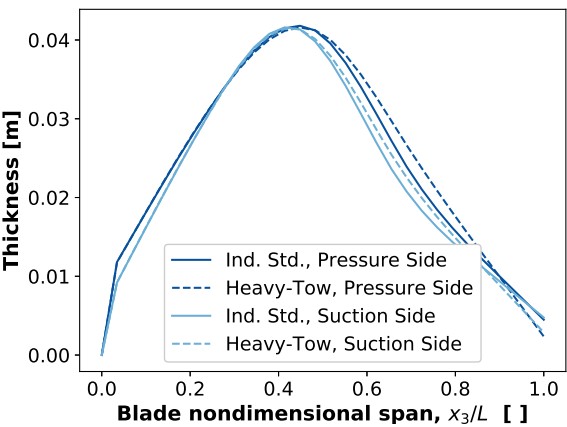

**Figure 14.** Comparison of optimized spar cap thickness distribution for the BAR-DRC design using industry-standard carbon fiber unidirectional composite and heavy-tow fiber composite.

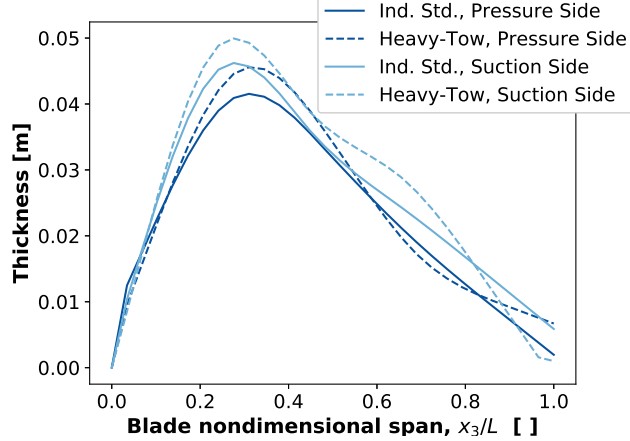

**Figure 15.** Comparison of optimized spar cap thickness distribution for the BAR-USC design using industry-standard carbon fiber unidirectional composite and heavy-tow fiber composite.

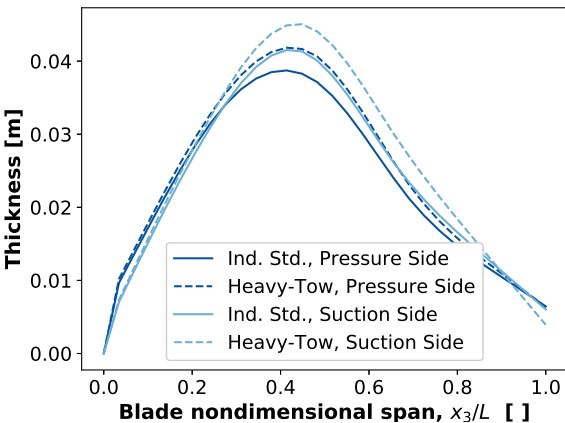

**Figure 16.** Comparison of optimized spar cap thickness distribution for BAR-URC design using industry-standard carbon unidirectional composite and heavy-tow fiber composite.

## 6 Conclusions

Long and flexible blades were designed in a multifidelity optimization collaboration between NREL and SNL. NREL focused on minimizing the levelized cost of energy of each design with a system-level optimization. Since beam FE models of the blades were utilized at the system level, higher structural fidelity was utilized by SNL to minimize mass while simultaneously
producing realistic design. Numerous updates to NuMAD were performed, which included an object-oriented approach with new capabilities comprising yaml file compatibility, optimization, and new structural analyses.

Blades that are transportable via controlled flapwise bending were found to be a viable approach to alleviate the logistical issues of oversized blades. For all designs with deflection and fatigue constraints, the deflection constraint was active and fatigue damage was inactive. Other constraints such as buckling and rupture were also found to be active; further confirming the need
for structural fidelity greater than a beam FE model. The results confirm that blade mass can, in general, be reduced substantially by going to a downwind design or by increasing nacelle tilt and rotor precone angles above conventional values. The downwind designs were found to be dominated by sudden, abrupt events like gusts, rather than normal operating conditions. If downwind designs are pursued, load-alleviation strategies, such as bend-twist coupling, and active aerodynamics are recommended for future research. The use of the heavy-tow carbon in the spar caps was found to yield a 9-13 % cost savings compared to an
industry-standard carbon fiber.

*Code and data availability.* The NuMAD code as well the yaml files and other inputs are publicly available at https://github.com/sandialabs.



*Author contributions.* JP and NJ performed conceptualization, funding aquisition, project administration, and supervision. EC and EA implemented the new NuMAD methodology and prepared the original draft. PB and RF assisted with the BeamDyn investigation and PB assisted with the conceptualization and cost analysis. All authors have performed editing and review.

*Competing interests.* The authors declare that they have no competing interests in executing and publishing this work.

*Acknowledgements.* Sandia National Laboratories is a multimission laboratory managed and operated by National Technology & Engineering Solutions of Sandia, LLC, a wholly owned subsidiary of Honeywell International Inc., for the U.S. Department of Energy's National Nuclear Security Administration. This manuscript has been authored by Sandia National Laboratories under Contract No. DE-NA0003525 with the U.S. Department of Energy (DOE) and by the National Renewable Energy Laboratory, operated by Alliance for Sustainable Energy, LLC, for the U.S. DOE under Contract No. DE-AC36-08GO28308. Funding was provided by the U.S. DOE Office of Energy Efficiency and Renewable Energy Wind Energy Technologies Office. The views expressed in the article do not necessarily represent the views of the DOE or the U.S. Government. The authors would also like to acknowledge Chris Kelley, Brandon Ennis, and Ryan Clarke for their meaningful contributions to the project.



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
