# Peer review of "Land-based wind turbines with flexible rail transportable blades – Part II: 3D FEM design optimization of the rotor blades"

_Wind Energy Science, 2021_

## Author Response (AR1)

**REVISION TO MANUSCRIPT DRAFT**

**Journal** Wind Energy Science
**Manuscript ID** WES-2021-74
**Title** "Land-based wind turbines with flexible rail transportable blades – Part II: 3D FEM design optimization of the rotor blades"

The authors would like to thank the associate editor Prof. Sandrine Aubrun and the two reviewers for their time and valuable feedback. We believe that these inputs have contributed to the improvement of the paper as well as increasing the author's knowledge of the field in general. Below you will find a list of point-by-point replies to the reviewers' comments. The text from the reviewers is reported in italics and has been divided into a numbered list. Each point is followed by the authors' reply.

**Reviewer #1**

*[Reviewer] This is not the easiest paper to review because of the large amount of information contained. In my opinion, the paper is very well written, the analysis provided is highly valuable for the wind energy community and the contribution is original.*

*However, although I know the field, I am not expert in structural design of wind turbine blades. This is why, first I recommend the editor to incorporate to the review process experts in the field of blade structural design analysis and its optimization, who really will provide a more specific sound advice on the acceptance of the paper.*

*Considering my background, I think I can contribute with some general comments that could clarify some points of the study for the wind energy community.*

1. *The authors propose a detailed FEM analysis of different structural design options. It is not clear if their designs, that on the other hand lead to very flexible blades, can present deformation of the blade sections shapes that could affect the performance of the airfoils. A comment on this aspect could be convenient, because if this is the case, then a simple Blade Element Momentum approach (such as the one implemented OpenFast) will not be enough for determining the aerodynamic loads.*

   **[Authors]** We thank the reviewer for their time and valuable feedback; especially since structural design of wind turbine blades is outside of their expertise. We find the issue of deformations of the cross sections during operation to be very intriguing and we plan to investigate it further in the coming years. Current frameworks to run the load analysis meet the tradeoff between computational efficiency and fidelity by assuming that the outer shape of the blade is fixed. Airfoil performance is described in terms of polars, and these do not depend on the deformed shape of the blade. Overcoming this simplification requires a full fluid-structure interaction approach coupling a 3D blade-resolved computational fluid dynamics framework to a 3D finite-element solver. Literature offers various examples of such frameworks, see for example DOIs 10.1002/fld.2454 and 10.1016/j.jfluidstructs.2019.03.023, but standard design load cases described in the international standards cannot be run within these frameworks due to excessive computational costs and fragility of the implementations. Overall, in this work we decided to put the emphasis on the structural aspects of the blade designs. These assumptions and considerations are now more clearly pointed out in Section 3.

2. *[Reviewer] One general doubt is if there is any verification of the impact of the different blade designs on the power production of a wind turbine using the different blade options. Again a comment on this aspect could be convenient.*

   **[Authors]** As discussed in detail in Part 1 of this study ([https://wes.copernicus.org/articles/6/1277/2021/wes-6-1277-2021.pdf](https://wes.copernicus.org/articles/6/1277/2021/wes-6-1277-2021.pdf)), the downwind

rotors were found to generate less power due to decreased cut-out speed and reduced rotor-swept areas due to loading. We agree with the reviewer that a comment on this aspect should be better highlighted in the present article and a note was added to Section 5.2.

3. **[Reviewer]** *Commenting on how the different blades (different structural designs) will interact with the rest the wind turbine, or at least a comment clarifying that this aspect is relevant and will be treated in future research, would be convenient.*

   **[Authors]** We agree with the reviewer that these aspects are important. The response to this comment is grouped into the response to the next comment (#4) since it also deals with interactions with the rest of the turbine.

4. **[Reviewer]** *In general, some comments should be included on the necessity of checking the influence of the different proposed design options on the aeroelastic behaviour of whole wind turbine and in particular the interaction with the tower dynamics and the implications related to the control of the wind turbine.*

   **[Authors]** We agree with the reviewer that these aspects are important. The aeroelastic performance of the turbine designs is described in the companion paper, Part I. A more detailed aeroelastic analysis in the frequency domain is actively being conducted in the second phase of the Big Adaptive Rotor (BAR) project, focusing on system dynamics and aeroelastic instabilities. During the second phase of BAR we will also redesign the towers for each individual design, exploring tradeoffs and opportunities for cost savings. The impact of the location of the center of gravity of the rotor-nacelle-assembly will be investigated, especially for downwind rotors where aerodynamic thrust and rotor gravity loads align and generate larger moments along the tower compared to equivalent upwind configurations. Improved design detail of the controller will also be addressed in Phase 2 through controls co-design. These research items are now listed in a paragraph about future work in Section 6.

**Reviewer #2**

**[Reviewer]** *This paper is part of an interesting piece of research that investigates both on a system level and a detailed blade structural design level. The paper is well written and offers a lot of information, but there also some aspects that can be elaborated on.*

1. **[Reviewer]** *Especially the implications of high flexibility necessary for rail transport in the USA will trigger the potential reader. In this paper however a 'rail transport' constraint is missing, which is a pity since it could illustrate the consequence of the lower design strain for the Heavy-Tow carbon compared to the uniaxial glass or the industry-standard carbon fiber.*

   **[Authors]** We thank the reviewer for their time and insightful feedback. We agree that the explicit rail transport constraint is missing from the present analysis. We do, however,

have a proxy constraint by equating the tip deflection from this analysis to the low-fidelity analysis. Recall, that the low-fidelity optimization in Part 1 does explicitly enforce the rail-transport constraint. Thus, the rail-transport constraint is satisfied in the present analysis through iterations with the low-fidelity optimization. A note with this regard was added to Section 4.1.

2. *The reader could now conclude from Part 1 of this study (see wes-2021-29) that for each material a maximum strain of 3500 μstrain has been allowed. That would be low for uniaxial glass but high for the Heavy-tow carbon fiber.*

   **[Authors]** We thank you for pointing this out. In Part 1, the 3500 microstrain limit was selected based on the authors' expectation that other failures (e.g. adhesive) will occur after 3500 microstrain in the sparcap is reached. This results in a conservative design for the glass spar caps as you mentioned and acceptable conservatism for the heavy-tow spar cap; which has a factored rupture strain of 4200 microstrain ($\frac{\gamma_\sigma^{-1} X'}{E_1^*} = \frac{0.674\ \text{GPa}}{160.6\ \text{GPa}} \frac{10^6 \mu\varepsilon}{[\text{m/m}]} = 4197\mu\varepsilon$).

3. **[Reviewer]** *The blade design is discussed in Ch. 4.3 only briefly. It would help the reader to learn more about the choices made regarding the components. To name some questions that could be answered:*

   - *Has carbon fiber also been used for the LE and TE reinforcements?*
   - *The TE panel is a sandwich with triax facings, has that also been used for he LE panels?*
   - *Usually, the root section of a blade mainly consists of triax material, has that been used here too (Figures 14-16 do suggest something like that)?*

   **[Authors]** We agree that the reader can be aided by further clarification on this matter. The manuscript has been updated to reflect further detail of the construction of each component. See Section 4.3 and Table 4 in the revised manuscript.

4. **[Reviewer]** *Some notes on the ANSYS analysis. For the ANSYS model shell181 has been used. Shell elements like shell181 are suspected to give an incorrect estimate of the torsional stiffness and thereby of the shear stresses due to torsion (and maybe also shear forces). Would an inaccuracy of 30% in torsional (shear) stress lead to another ranking in blade designs?*

   Indeed, numerical [1] and experimental [2-3] evidence has been shown that shell elements, such as ANSYS SHELL181, can provide an overly compliant torsional response. Such errors occur when the reference surface of the shell model coincides with the outer mold line (OML) of the blade. Both [2] and [3] show that angle of twists from a torsional load applied to such a shell model were about 30% higher than experimentally determined values. Since the present analysis uses offset shells, we can expect that any twisting angles present in the analyses are over predicted by nearly the same percentage. Fortunately, the angle of twist was not a needed quantity for evaluation of the constraints

as you might already know. It would affect the angle of attack seen by various airfoils along the blade and thus would alter the loads; however, loads were transferred from the low-fidelity system optimization to the present analysis. They were considered in the present work to be fixed for a given design and therefore the error would not affect the loads applied to the shell model.

Shear stresses due to a given torsional load is what you specifically asked for. Since shear stress is needed for evaluation of the material rupture constraint, errors in shear stress have the potential to alter blade rankings. However, no study could be found that quantified the shear stress error from offset shell models; only the angle of twist as mentioned above. It should be noted that a 30% error in the angle of twist need not correspond to a 30% error in shear stress due to torsion. An example of this decoupling is seen in various mesh convergence studies. For a displacement-based FE code, as you may know, the displacements, in general, converge faster than stresses do. Further study would be needed to quantify the effect of shell offset on shear stress. Since the primary deformation mode of these blades is flexural, we leave this to a future work.

The manuscript was thus modified in the following way. In Section 4.4 we state that we use SHELL181 elements with a reference surface that coincides with the OML. We also acknowledge that this has been observed to overpredict the torsional angle of twist by 30%. We defend the use of SHELL181 by stating our assumption that bending behavior is the only concern.

[1] Malcolm, D. J., & Laird, D. L. (2007). Extraction of equivalent beam properties from blade models. *Wind Energy: An International Journal for Progress and Applications in Wind Power Conversion Technology*, *10*(2), 135-157.

[2] Branner, K., Berring, P., Berggreen, C., & Knudsen, H. W. (2007, July). Torsional performance of wind turbine blades–Part II: Numerical validation. *In International conference on composite materials (ICCM-16)* (pp. 8-13).

[3] Fedorov, V. (2012). Bend-twist coupling effects in wind turbine blades. *DTU Wind Energy, Denmark.*

5. ***[Reviewer]*** *The Tsai-Wu criterion is applied, but line 257 mentions 'that neglects any effects due to transverse normal stress'. What is mentioned here: all transverse normal stresses, i.e. all non-spanwise normal stresses, or only the normal stress perpendicular to the laminate plane?*

   **[Authors]** Transverse normal stress, here, indicates normal stress perpendicular to the laminate plane. We thank the reviewer for catching this and have added a clarification in Section 4.1.

6. ***[Reviewer]*** *For the Heavy-Tow carbon fiber most probably material data are used for pultruded profiles. If that is the case the spar cap will have discrete thickness steps (the*

*pultrusion thickness), which will lead to a somewhat higher blade mass. At the thickness step a stress peak will occur which lowers the strength locally which again leads to an increase in blade mass. What consequences would this have?*

**[Authors]**  We agree that the heavy-tow material would likely have discrete thickness steps. NuMAD builds a stepwise model and fabricators can and do adjust the manufactured stepping resolution to correspond to the NuMAD spacing. That is what we have done in prior experimental programs, and the specimen was built as close as possible the NuMAD model. As for the effect of stress concentrations in the present blade designs. This effect is unknown to the authors since such localizations of stress cannot be captured with shell elements. A study with brick elements would be required for an accurate assessment. At this phase of the exploration of the new technology, we leave this aspect to future studies which can accommodate higher fidelities via a reduced design-space. Obviously, if the "as built" blade is heavier than the model this could affect the cost of the blade and could also increase edgewise bending moments.